# Elementary Students’ Perceptions of Cognitively Challenging Physical Activity Games in Physical Education

**DOI:** 10.3390/children9111738

**Published:** 2022-11-12

**Authors:** Athanasios Kolovelonis, Evdoxia Samara, Nikolaos Digelidis, Marios Goudas

**Affiliations:** Department of Physical Education & Sport Science, University of Thessaly, 421 00 Trikala, Greece

**Keywords:** qualitative research, cognitive development, physical education

## Abstract

This study examined 10–11-year-old students’ perceptions regarding three different types of physical activity games based on different principles of mental engagement (contextual interference, mental control, or discovery). A total of 156 students (84 girls) from five fourth-grade (75 students) and five fifth-grade (81 students) classes of five elementary schools located in a middle-sized city in central Greece participated in the study. These students participated in a larger project consisting of a series of acute experiments aiming to examine the effectiveness of cognitively challenging physical activity games in elementary physical education. Students responded to open-ended questions regarding their perceptions of the physical activity games. Their responses were analyzed through a thematic analysis. A total number of 706 quotes were identified and categorized into the lower-order themes which were organized into four higher-order themes: (a) characteristics of the games, (b) effects of the games, (c) areas for improvement, and (d) preferences for specific games. Students’ views provided supporting evidence regarding the employment of cognitively challenging physical activity games in physical education. Useful insights regarding the characteristics of the games, their effects, and their areas for improvement were also gained.

## 1. Introduction

Physical education can contribute to students’ psychomotor, cognitive, and social development in preparing a physically educated person. That is, individuals who are competent and feel confident in a variety of physical skills are motivated towards physical pursuits, participate regularly in physical activity, are physically fit, and value physical movement and activity and its contribution to a healthy lifestyle [1]. Childhood and pre-adolescence are considered sensitive periods for developing and mastering fundamental motor skills and introducing students to a wide variety of sport skills [2]. Moreover, one of the major aims of physical education is to promote students’ physical activity and health-related fitness [3]. Indeed, according to the World Health Organization [4], school physical education is an avenue for promoting students’ physical activity as the majority of students can be reached and provided appropriate interventions.

For achieving all these multiple goals, physical educators should adopt the most effective instructional approaches and select the most appropriate content for helping students to develop skills, knowledge, and attitudes required for a lifetime of physical activity [5]. Indeed, the content of physical education may include activities such as games and sports, fitness activities, dance and rhythmic activities, and individual performance activities [3]. This study focused on a certain type of physical education activity, the cognitively challenging physical activity games, to examine students’ perceptions and experiences regarding their participation in a physical education session including this type of game.

The cognitively challenging physical activity games not only involve students in physical activity and energy expenditure but also require them to be cognitively engaged in order to be successful in the game. In particular, these games engage students in novel, unpredictable, changing, and complex conditions, providing them with opportunities to adapt their movement patterns and to explore new and diversified responses or strategies for playing the game [6]. Moreover, the physical activity games require students to respond in movement patterns characterized by higher levels of coordinative difficulty and low levels of automaticity. These games trigger students’ cognitive involvement by altering the rules of the games to be more challenging, altering the roles of students during the games, and promoting problem solving and divergent discovery [7].

For designing cognitively challenging physical activity games, one should follow the three principles of mental engagement, namely, highlighting contextual interference, emphasizing mental control, and promoting discovery [6]. In particular, in contextual interference conditions the context and the conditions of a game change, requiring students to make unpredictable sequences of actions. Thus, for creating contextual interference the game should involve students in nonrepeating, random conditions, and the actions required by students should not be automatized but should change randomly. Under these game conditions, students learn better because they have to elaborate and think deeply about the changing features of the game. Moreover, mental effort is required to memorize the requirements of the various movement plans and to reconstruct and plan multiple mental operations and actions to respond to the changing conditions of the game.

Another characteristic of the cognitively challenging physical activity games is that they highlight mental control. Indeed, considering that the conditions of these games change often and are sometimes unpredictable, students are required to hold and manipulate information regarding the various aspects of the game including the appropriate movement patterns or the rules applied in each game condition. Moreover, mental control is triggered when during the game students stop what they are doing in order to act in a totally different way or to respond to a variety of signals which sometimes are contradictory. Promoting discovery is the third characteristic of the cognitively challenging physical activity games. For promoting discovery, the physical activity games involve problem-solving conditions that require students to produce multiple solutions to an open-ended movement problem. Moreover, discovery can be promoted through the use of open-ended games. In these games, various aspects of the game are set (e.g., the goal of the game, the starting point, the rules) and students should select the most appropriate actions or strategies in order to be successful.

The idea of using physical activity games for enhancing simultaneously students’ physical activity and cognitive development is supported by recent evidence suggesting a shift in the approach of designing physical activity programs [8,9]. This new approach calls for a shift “from simply moving to moving with thought” [10]. This can be realized by adopting a shift from focusing only in the “quantity” of physical activity to increasing the “quality” of physical activity and emphasizing improvements in multiple domains including the cognitive one [7,11]. That is, physical activity programs should promote health-related outcomes increasing students’ physical activity and enhancing at the same time their cognitive development [12,13]. Following this approach, the physical activity programs can “fill two needs with one deed” [14] by focusing on both cognitive and motor development [15].

Research evidence has shown that physical activity can have positive effects on students’ cognitive development [14,16,17]. However, in physical education only a few studies have examined the effects of physical activity on students’ cognitive development. For example, a 30 min aerobic exercise session [18] and a 60 min basketball-based activity [19] enhanced students’ executive functions. Moreover, a long-term intervention based on team sports (i.e., floorball and basketball) with a high level of physical exertion and cognitive engagement enhanced elementary students’ executive functions more than the aerobic program and the control condition [20]. Similarly, a 6-month physical education program including cognitively demanding activities in tennis resulted in higher inhibition both in fitter and overweight children [21].

This research evidence suggested that physical activity can enhance students’ cognitive development. However, further research in this field is required [22]. For example, one area that needs further investigation is the types of physical activity that can have the highest impact on students’ cognitive development. Indeed, a recent review and meta-analysis [16] suggested that physical activity interventions were not always effective in promoting students’ executive functions. Cognitively challenging physical activity games are a promising means for developing both students’ cognitive and physical development in physical education. Indeed, executive functions are optimally developed when students are involved in cognitively complex physical activity experiences [20] and novel, challenging, and diversified but not highly repetitive and automatized tasks [23]. Such conditions are included in the cognitively challenging physical activity games. Recently, [24] found that a single physical education session with cognitively challenging physical activity games had positive effects on experimental group students’ executive functions. The three different types of physical activity games involved in this study, which were each based on a different principle of mental engagement (contextual interference, mental control, or discovery), were equally effective in triggering students’ executive functions. Similarly, students who were involved in a single physical education session with such games improved their scores in the executive functions more than students who were taught soccer or track and field skills [25] or students who were involved in traditional activities for enhancing their health-related fitness components [26]. Moreover, a four-week intervention in physical education involving cognitively challenging physical activity games had positive effects on students’ executive functions [27].

This preliminary evidence suggests that introducing cognitively challenging physical activity games in physical education may have positive effects. However, further research is needed in this field. Most importantly, students’ views regarding these types of games should be considered. For promoting students’ participation in physical education, the most appropriate and appealing activities should be selected. A recent study showed that boring content delivery, along with a focus on competition and the misuse of fitness tests, were the main reasons for declining student attitudes towards physical education [28]. Indeed, evidence has suggested that repeated exposure to the same activities or stimulus may decrease students’ interest and enjoyment [29]. This may have detrimental effects on students’ motivation to be involved in physical activity because having fun has been perceived by students as one of the main motives and benefits of participating in physical activity [30]. This may be more evident for girls, as their attitudes towards physical education [31] and their enjoyment seems to worsen compared to boys, especially when it is combined with lower perceived athletic competence [32]. Although some evidence suggests that girls may prefer individual and expressive activities, further research is needed regarding the specific types of meaningful experiences in physical education for boys and girls [33]. The use of novel, complex, and cognitively challenging activities may be an alternative [29]. The physical activity games fall into this category, and their use may have positive effects for students’ physical and cognitive development. Indeed, fun, challenging, socially interactive, and personally relevant learning activities are the basis for designing meaningful physical education experiences for students [33].

Thus, to enrich our knowledge regarding cognitively challenging physical activity games, a qualitative methodological approach was applied involving students responding to open-ended questions. This approach enabled students to express their perceptions and beliefs about aspects of the physical activity games and thus provide evidence regarding the effectiveness of these games and the appropriateness of their introduction in school physical education. In this way, the process of designing and implementing effective, meaningful, and enjoyable physical education programs will be facilitated. Moreover, potential underlying mechanisms behind the effectiveness of the physical activity games may be revealed, opening new areas for research in this field.

The aim of this study was to explore students’ perceptions regarding the cognitively challenging physical activity games they played during a physical education session. In particular, this study aimed at exploring whether students: (a) considered these games as new and novel and (b) perceived that they had learnt something new through playing these physical activity games.

## 2. Materials and Methods

The present study was part of a larger project aiming to examine the effectiveness of using cognitively challenging physical activity games in elementary physical education. The project consisted of a series of acute experiments that involved students in a single physical education session including physical activity games [24,25,26]. Students who participated in those acute experiments completed additional measures after the end of the physical education session and reported their perceptions regarding the physical activity games they played during the physical education session.

### 2.1. Participants and Settings

Participants were 156 students (*M*age = 9.86, *SD* = 0.62, 72 boys, 84 girls) from five fourth-grade (75 students) and five fifth-grade (81 students) classes of five elementary schools. Elementary schools recruited for this study were typical civil schools located in a middle-sized city in central Greece. Physical education in Greece is coeducational, mandatory, and delivered by specialized physical education teachers in three 45 min sessions per week for fourth-grade students and two 45 min sessions per week for fifth-grade students. The physical education curriculum for grades 4 and 5 includes the main team sports (i.e., basketball, volleyball, soccer, and handball), individual sports (i.e., track and field, gymnastics) and traditional dance.

### 2.2. Description of the Physical Activity Games

The 45 min physical education session consisted of 5–6 physical activity games that required energy expenditure but also involved students in cognitively demanding conditions including novel, challenging, and diversified but not highly automatized tasks [23]. These games challenged students by increasing the coordinative difficulty of movement patterns; altering the rules to make the game more challenging; altering the roles of students during the games, sometimes in an unpredictable way, to maximize effort; and encouraging problem solving and divergent discovery [7]. In particular, the physical activity games were designed based on the three principles of cognitive engagement, namely, contextual interference, mental control, and discovery [6]. For creating contextual interference, the conditions of the game change continuously, requiring students to respond with unpredictable sequences of actions. For highlighting mental control, the physical activity games involved students in conditions setting memory demands for holding and manipulating information, which included requiring students to stop what they were doing and to act in a totally different way or to override prior actions upon alternating signals to go and stop. Some of the physical activity games focused on promoting students’ discovery. These games involved problem-solving conditions or open-ended games requiring students to produce multiple and unique solutions or to select the most appropriate actions or strategies in order to be successful. An example of a game that students played during physical education sessions is hop, pop, and tag which is a tag game played in open space in which any student can tag any other student. The tagged students squat down and can return to the game when their tagger is tagged by another student. This game is different from traditional tag games. It challenges students to avoid being tagged by multiple taggers while they attempt to tag the other students. Moreover, tagged students must be aware if their taggers are tagged in order to return to the game [6].

### 2.3. Procedures and Data Collection

Ethical approval for the study was granted by the University Ethics Review Committee and the Ministry of Education. Permissions were also obtained from the school principals and the physical education teachers. Students participated voluntarily after written parental consent was obtained, and they were told that the purpose of the study was to test the effectiveness of some approaches in teaching physical education. Acute experiments including a physical education session with physical activity games were conducted in schools’ open sport facilities during the regular physical education schedule. They were implemented by an experimenter blind to the aims of the study and experienced in delivering physical education interventions. After participating in the physical education session with the physical activity games, students reported their views regarding these games by responding to the following four open-ended questions:What do you think about all the games and the activities you were involved in during the physical education session today?Did you learn something new today in the physical education session? If yes, please describe what that was.Did you do something different today in the physical education session compared to other physical education sessions? If yes, please describe what that was.Was there anything in the physical education session today that you did not like or that you would prefer to have been done in a different way? If yes, please describe what that was.

### 2.4. Data Analysis

Students’ responses were analyzed after completion of data collection through a thematic analysis [34,35]. Two of the authors read students’ responses and identified lower-order themes. Next, higher-order themes were identified inductively. Research questions guided this process. During the process of identifying the higher- and lower-order themes, the two authors discussed all the themes to reach a consensus. After that, the two authors independently categorized the raw information into the lower-order themes. The consistency between coders was high (93% of the quotes were categorized in the same theme). For the cases of quotes categorized in a different theme, the two authors discussed to reach a consensus. For enhancing the reliability of themes and the coding process, a third author provided comments and suggestions facilitating the process of analyzing the data and reaching the final agreements. To achieve trustworthiness [36], the experimenter familiarized herself with the school settings and the students while she kept notes of personal thoughts, nonverbal forms of communication during the physical education session, and any other issues related to the implementation of the physical activity games and students’ reactions to these games. Moreover, the entire process was monitored by a researcher experienced in qualitative methods who provided guidance and assistance in resolving any issue that arose and any cases of discrepancy or conflict. The triangulation of the data analysis was achieved by examining the notes of the experimenter regarding her personal thoughts and students’ reactions during the physical activity games, including nonverbal forms of communication during the physical education session, and students’ data regarding situational interest included in a battery of quantitative measures of the largest project. The experimenter’s notes regarding the settings for implementing the physical activity games (e.g., the climate in physical education lessons) were also considered [28].

## 3. Results and Discussion

The analysis of the data produced the thematic structure shown in Table 1. Four higher-order themes were identified: (a) characteristics of the games, (b) effects of the games, (c) areas for improvement, and (d) preferences for specific games. The high-order theme “characteristics of the games” referred to students’ perceptions about the characteristics of the cognitively challenging games. The theme “effects of the games” referred to students’ perceptions about potential learning outcomes that resulted from their participation in the cognitively challenging games. The theme “areas of improving” referred to potential actions for improving these games, while the theme of “preferences for specific games” included students’ preferences regarding specific games. Lower-order themes were also identified within these higher-order themes. A total number of 706 quotes were categorized in these lower-order themes. The frequencies of quotes in each lower-order theme were calculated for the total sample and separately for grade and gender, and they are presented in Table 1. Subtotals of the frequencies for each theme and subtheme were also calculated. All these results are presented and discussed next in separate sections for each higher-order theme. The discussion of the findings has been incorporated into the results section to help the reader place the findings and their interpretations within the existing related literature.

### 3.1. Characteristics of the Games

Almost half of the quotes from students’ answers were coded in the first higher-order theme, the characteristics of the games. The large majority of these quotes referred to positive characteristics of the physical activity games while a small number of quotes could be considered as negative. In particular, students found the physical activity games interesting, fun, and enjoyable. They liked the games they played; they found them very good or even perfect and reported that they played some of these games for the first time. Some characteristic quotes from students’ answers were the following:

“I liked the games a lot. I really enjoyed playing these games.”

“These were fun and interesting games.”

“They were fun, because we did something different.”

“These games were perfect.”

“They were perfect. I would like play these games again.”

“These games were very interesting and liked them. They were different from those we usually do in physical education.”

“I found them very interesting. I had a great time.”

These results suggest that the physical activity games were an engaging, fun, and appealing means for involving students in physical activity. This is important, because having fun has been perceived by students as one of the main motives and benefits of participating in physical activity [30,37]. Indeed, the role of enjoyment and interest in increasing students’ motivation for participating in physical education are widely acknowledged [38,39]. For example, students’ motivation for participating in physical education is associated with positive outcomes including increased levels of physical activity out of school [40], while enjoyment has been considered as an important mediating factor in motivating adolescents to be physically active [41]. Therefore, physical education should include a wide range of enjoyable and interesting activities and games [42], such as cognitively challenging physical activity games, in order to increase students’ motivation to participate. An interesting variation was that girls, in comparison to boys, reported a higher number of quotes indicating that the games were fun, enjoyable, and interesting. Thus, involving girls in cognitively challenging physical activity games may increase their positive experiences during physical education which in turn may positively affect their willingness to be involved in physical activity [43]. Given that girls have generally lower positive attitudes toward physical education than boys [44,45], this finding points out a potential avenue to tackle this issue.

Another important characteristic highlighted by students was the novel aspect of the physical activity games. Indeed, these games adopted the new approach of combining both physical and mental demands for providing students with opportunities for “moving with thought” [10]. This combination of the physical and cognitive demands may be perceived by students as something novel. Recent evidence has suggested that the satisfaction of novelty was the stronger predictor of intrinsic motivation for learning while novelty was also associated with vitality, dispositional flow, and satisfaction with physical education classes [46]. In contrast, a lack of novelty in exercise routines has been suggested as a key factor contributing to low physical activity participation [47]. Indeed, the repeated exposure to the same activities or stimulus may decrease students’ interest and enjoyment [29]. Thus, the cognitively challenging physical activity games can be used in physical education as an alternative to traditional physical activities, thus involving students in novel and complex activities promoting both their physical activity and cognitive development [12,13].

On the other hand, students reported a few negative comments about the physical activity games. However, the negative comments were few and represented only 12% of the quotes coded in the high-order theme of the characteristics of the games. Half of these negative quotes were students’ comments that they have played these games before, while a few students reported that they generally did not like the games or that they were boring or tiring. Some examples of quotes representing students’ negative perceptions regarding the games are the following:

“… I have played these games before.”

“I did not like them at all.”

“I found these games a little bit boring.”

“… it was, just, a little tiring.”

The fact that the number of negative comments regarding the physical activity games was very low is encouraging. However, physical education should involve all students to help them reach their potential, paying special attention to students who struggle with their participation in physical education. Therefore, future research should further explore potential negative aspects of the games, focusing on variables related to the learning environment but also on variables related to students’ personal characteristics (e.g., self-efficacy and preferences for sport activities). For example, three of the four negative comments were by fifth-grade students. This may suggest that games should become more challenging and demanding for older students.

### 3.2. Effects of the Games

The second largest higher-order theme, in terms of the number of quotes coded, was the effects of the physical activity games. Two out of three quotes coded in this higher-order theme represented positive learning outcomes while the remaining one-third indicated no learning effects. Students reported that they learnt to play new games, while some quotes referred to learning in general. Most importantly, students reported that through the physical activity games they learnt specific strategies and skills such as to cooperate with their teammates and to trust them, to concentrate on what they were doing, to think better, to persist in their efforts, and to play fair. Some representative quotes indicating students’ learning outcomes through their participation in the physical activity games are the following:

“I learnt new and interesting games.”

“I learnt some new games, such as that of traffic lights and the cars.”

“I learnt a lot of things in this physical education session.”

“… how to cooperate and to be closed with my classmates.”

“I learnt to never give up.”

“… to concentrate more.”

“I learnt how to trust my friend.”

Learning new games is an important outcome in physical education. Being involved in new games may help students to develop motor and sport skills and to feel competent and confident, thus facilitating the process of becoming a physically educated person [1]. Indeed, an enriched intervention in physical education focusing on physical fitness, motor coordination, cognition, and life skills, and including physical activity games, had positive effects on students’ motor coordination skills (e.g., ball skills and static/dynamic balance) [22]. Students’ perceptions of learning new games may also be associated with their motivation. Indeed, introducing new content (i.e., new physical activity games) can attract students’ interest and increase their motivation to participate in physical education [46], providing them with opportunities to move with thought [8,10] and to learn while having fun [6].

Moreover, some students reported that through the physical activity games they learnt to concentrate and persist. These skills are associated with positive outcomes in physical education, such as enhanced performance [48]. Moreover, other students reported cooperation and fair play as learning outcomes. This may be the result of the positive learning environment that physical activity games promote by focusing on involving all students in cognitively challenging conditions. Indeed, research in physical education has shown positive relationships between an autonomy-supporting motivational climate and students’ perceptions of positive social behavior including cooperative skills [49]. These skills can also be considered as life skills, that is, skills that students can learn in physical education and then transfer to other life domains.

Some students reported that they did not learn something new from their participation in the physical education lesson. The most representative quotes in this subtheme are the following:

“… no, I did not learn something new.”

“I did not learn something because I knew these games.”

Some of these students only reported that they did not learn anything, while some others reported that they already knew the games. An explanation for this result may be the fact that some of the physical activity games were based on already known games which were modified to become more challenging for students. For example, hop, pop, and tag (see the method section for the description) is a modified tag game setting more challenges to students’ memory and cognition compared to regular tag games. Probably, some students may focus on the general pattern of the game without discerning its unique aspects. However, this interpretation should be further explored in future research.

### 3.3. Areas for Improvement

Students provided comments regarding potential changes they would make in the physical education session. Almost two-thirds of the students made comments indicating they would make no changes to the physical activity games they played or the way of delivering them. Some representative quotes are the following:

“I would not change anything. I liked the way we played the game and the game itself.”

“The session was perfect. I would not be in a different way.”

These results suggest that the majority of students were generally satisfied with their participation in these physical activity games and thus they would not change anything. This is important, because increased levels of satisfaction during physical education are associated with increased satisfaction for physical activity outside of school [50].

On the other hand, some comments including suggestions and potential changes were also made. These included students’ suggestions for modifying games, students’ willingness to play the games again, and suggestions to play more games, to play some specific games, or to play some other sport or physical activity. Some representative quotes from students’ answers are the following:

“In the last game, I would like to have more numbers in order to find out who can remember them.”

“I like them all and I would like to play them again.”

“… to play again the game see, sky, and earth.”

“I would like to play more games, especially educational games.”

“I liked them, but I would like to play soccer.”

Students’ suggestions for improving the games may be a valuable source of information that could be used for making these games more effective. Most importantly, physical educators may involve their students to modify the physical activity games in order to make them more challenging and appealing. Providing students with instructional choices can enhance their autonomy and engagement in school physical education [45]. Moreover, a few students reported that they would prefer to play a different sport (e.g., soccer). This may be explained by the fact that the physical education curriculum in Greece is centered on teaching team sport, such as soccer, volleyball, and basketball, and thus many students are used to playing these team sports during physical education.

### 3.4. Preferences for Specific Games

Students reported their preferences for specific physical activity games that they liked or did not like. This higher-order theme included a lower number of quotes compared to the other higher-order themes. Almost all the quotes referred to students’ preference for games they liked. Some examples of students’ quotes are the following:

“I think that all games were good, but I liked most that with the mirror.”

“… especially, I liked that with the numbers and the handkerchief.”

“I did not like the game with the pieces of paper. It was a little boring.”

Involving students in games they like can increase their fun and enjoyment and thus their motivation for participation in physical education [38,39]. Moreover, taking into consideration students’ preferences for specific games may enhance their feelings of ownership in physical education which in turn has been found to be associated with their satisfaction from their participation in physical education [50].

From an applied perspective, the results of this study provided information that could be considered in the design and implementation of physical education programs. The physical activity games are viewed by students as an enjoyable, fun, and novel means of being involved in physical activity during physical education. Thus, physical educators should use such games to provide students with opportunities to get involved in physical activity while at the same time being engaged in cognitively challenging conditions and having fun [51,52]. Moreover, physical educators may involve students in the process of selecting the games they play to increase their feelings of autonomy and ownership and thus to enhance their motivation for participating in physical education. Students should also be encouraged to modify these games to make them more challenging and engaging.

This study adopted a qualitative approach for examining students’ perceptions regarding the physical activity games. Students provided written responses to four open-ended questions, reporting their experiences about their participation in the physical education session including physical activity games. This approach had the advantage of involving a large number of students who expressed their views on the physical activity games. However, this approach did not permit follow-up questions to obtain explanations or to look more deeply into students’ perceptions. Thus, future research should involve in-depth interviews to further explore students’ views regarding the physical activity games. A combination of these two approaches may also be used. That is, in the first phase all students who participate in the physical activity games would respond to the open-ended questions, while in the second phase some of these students would be selected to be involved in in-depth interviews.

## 4. Conclusions

This study adopted a qualitative research method and gained useful insights regarding students’ views on cognitively challenging physical activity games. This information can be used for further developing these kinds of games and designing and implementing effective physical education interventions. In particular, physical activity games are considered appropriate content for motivating students to be involved in physical activity during physical education. Indeed, these games are designed to promote physical activity and energy expenditure while maximizing enjoyment and fun and involving students in cognitively demanding conditions [6,11]. Cognitively demanding conditions are created when students are involved in novel, unpredictable, random, and complex game conditions, including tasks with higher levels of coordinative difficulty and low levels of automaticity and tasks and games promoting problem solving and divergent discovery [6,7]. Thus, considering that physical education targets multiple goals, the cognitively challenging physical activity games can be considered appropriate content for enhancing students’ psychomotor, cognitive, and social development.

## Figures and Tables

**Table 1 children-09-01738-t001:** Thematic structure of the data and frequencies of the quotes.

	Number of Quotes				
	Total	Grade		Gender	
		4th	5th	Boys	Girls
**Theme 1: Games’ characteristics**	** *317* **	** *125* **	** *193* **	** *129* **	** *188* **
Positive	** *271* **	** *113* **	** *158* **	** *109* **	** *162* **
Interesting	38	11	27	16	22
Fun, enjoyable	119	54	65	47	72
Perfect/very good	52	23	29	19	33
Novel (new/different)	59	25	34	25	34
Easy to play	3	0	3	2	1
Negative	** *46* **	** *12* **	** *35* **	** *20* **	** *26* **
Boring	5	1	5	3	2
Tiring	6	0	6	1	5
Common/played before	23	7	16	8	15
Did not like (general)	9	4	5	7	2
Other	3	0	3	1	2
**Theme 2: Games’ effects**	** *178* **	** *81* **	** *97* **	** *90* **	** *88* **
No learning	60	25	35	32	28
Learning outcomes	** *118* **	** *56* **	** *62* **	** *58* **	** *60* **
Learning new games	61	31	30	31	30
General learning	21	12	9	6	15
Exercise and physical activity	8	2	6	7	1
Strategies and skills	** *28* **	** *11* **	** *17* **	** *14* **	** *14* **
Cooperation	10	5	5	3	7
Concentration	8	2	6	6	2
Other	10	4	6	5	5
**Theme 3: Areas for improvement**	** *132* **	** *57* **	** *75* **	** *59* **	** *73* **
No changes	93	43	50	40	53
Changes and suggestions	** *39* **	** *14* **	** *25* **	** *19* **	** *20* **
To play the games again	17	7	10	6	11
To play more games	2	1	1	0	2
To play specific games	4	2	2	2	2
To play other sports/activities	5	2	3	3	2
Suggestions for modifying games	11	2	9	8	3
**Theme 4: Preferences for specific games**	** *73* **	** *34* **	** *39* **	** *29* **	** *44* **
Games students liked	64	32	32	24	40
Games students did not like	9	2	7	5	4
Theme 5: Other	6	3	3	3	3

*Note*: Number in italics and bold represent subtotals for the respective category.

## Data Availability

Not applicable.

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
