# Peer review of "Elementary Students’ Perceptions of Cognitively Challenging Physical Activity Games in Physical Education"

_children, 2022, doi:10.3390/children9111738_

Round 1

Reviewer 1 Report

in the abstract:  This study examined students' perceptions regarding three different types of physical activity games based on different principles of mental engagement (contextual interference, mind control, or discovery). It should indicate the age of the students

The presentation of the topics and the quotes are also timely.

It would have been interesting to see a network showing the relationships between quotes, a cloud of words or a map, but I leave it to the freedom of the authors

01

Author Response

Thank you for reviewing our manuscript and for providing comments for improving its quality. We have now revised our manuscript addressing and responding to all your comments. Please see our detailed responses below. Revisions in the text have been highlighted in the manuscript in the form of “track changes”.

Comments and Suggestions for Authors

in the abstract:  This study examined students' perceptions regarding three different types of physical activity games based on different principles of mental engagement (contextual interference, mind control, or discovery). It should indicate the age of the students

Following your suggestion, we have added the age of students in the abstract.

The presentation of the topics and the quotes are also timely.

 Thank you for this positive evaluation

It would have been interesting to see a network showing the relationships between quotes, a cloud of words or a map, but I leave it to the freedom of the authors

Respectfully, we have not followed your suggestion, since a graphical representation of the themes would not add information to the reader additionally to that presented in the Table.

Reviewer 2 Report

I begin by thanking you for the opportunity to review a paper that is interesting from a pedagogical point of view.

Despite the clarity in the contextualization of the study and the presentation of the results, some aspects need to be improved, in particular:

11. The introduction should include more and more recent studies on students' perspectives on physical education (they only mobilize two studies, one of which is over fifteen years old).

22. Still in the introduction, it was interesting to present studies showing whether or not there are differences in the way boys and girls look at physical education activities, considering that they have data that allow making this analysis.

33. Following point 1., the Introduction should generally be revised, seeking to include more recent bibliography; of the 46 references presented, only 24% are from the last 5 years.

44. In the data analysis it would be important to clarify what defines/characterizes each of the higher-order themes.

  5. Considering the data presented in table 1 (pp. 5-6), it would be interesting to carry out an analysis of the data for boys and girls.

66. The conclusions should be deepened, focusing on particular aspects that the type of activities proposed should consider indispensable in order to meet the objectives of being cognitively challenging activities.

Author Response

Thank you for reviewing our manuscript and for providing us with constructive comments.

We have now revised our manuscript addressing and responding to all your comments. Please see our detailed responses to each specific comment below.

Accordingly, we have also highlighted the respective edits in the manuscript.

Comments and Suggestions for Authors

I begin by thanking you for the opportunity to review a paper that is interesting from a pedagogical point of view.

Thank you for reviewing our manuscript and for this positive evaluation.

Despite the clarity in the contextualization of the study and the presentation of the results, some aspects need to be improved, in particular:

  1. The introduction should include more and more recent studies on students' perspectives on physical education (they only mobilize two studies, one of which is over fifteen years old).

Thank you for this constructive comment. Following your suggestion, we have added more recent studies regarding students’ perspectives on physical education (lines 85-88, 134-137).

  1. Still in the introduction, it was interesting to present studies showing whether or not there are differences in the way boys and girls look at physical education activities, considering that they have data that allow making this analysis.

Following your suggestion, we have commented in the introduction about the gender differences in attitudes and motivation towards physical education and the potential gender differences regarding physical education experiences (lines 141 - 152).

  1. Following point 1., the Introduction should generally be revised, seeking to include more recent bibliography; of the 46 references presented, only 24% are from the last 5 years.

Thank you for this constructive comment. Following your suggestion, we have revised the introduction to include more recent research studies (for example, lines 86-87, 97, 122-129, 137, 142, 147, ).

  1. In the data analysis it would be important to clarify what defines/characterizes each of the higher-order themes.

Following your suggestion, in the revised manuscript we have added a description of each of the higher-order themes (lines 260 - 266). 

  1. Considering the data presented in table 1 (pp. 5-6), it would be interesting to carry out an analysis of the data for boys and girls.

Respectfully, we have not followed your suggestion, because a statistical analysis would not be akin to the concept of the study, that is to elicit students perception through open-ended questions. Besides, the only notable difference in the responses of boys and girls was on the first higher order theme (Games’ Characteristics) and especially on the positive comments. We had commented on this difference and its implication in the initial submission (lines 306 - 309) and we added further materials (lines 310 - 312).

  1. The conclusions should be deepened, focusing on particular aspects that the type of activities proposed should consider indispensable in order to meet the objectives of being cognitively challenging activities.

Following your suggestion, we have revised the conclusion section to highlight the nature and the characteristics that tasks and games should have in order to be characterized as cognitively challenging activities (lines 481 - 485).

Round 2

Reviewer 2 Report

I thank the authors for clarifying and improving the quality of the paper. I believe that it meets all the scientific conditions to be published.